# Predicting Healthcare Workload in Pediatric Home Hospitalization: The Role of Patient Complexity and Family Participation

**DOI:** 10.3390/healthcare13233066

**Published:** 2025-11-26

**Authors:** Astrid Batlle, Andrea Pardo, Pepus Daunis-i-Estadella, Raquel García Romero, Sandra López-Mateo, Ane Achotegui, Elisenda Esquerdo, Carmen Villalón, Anna Marín, Mariona Fernández de Sevilla, Andrea Aldemira

**Affiliations:** 1Home Hospitalization Service, Hospital Sant Joan de Déu, 08950 Barcelona, Spain; andrea.pardo@sjd.es (A.P.); raquel.garciar@sjd.es (R.G.R.); sandra.lopezm@sjd.es (S.L.-M.); ane.achotegui@sjd.es (A.A.); elisenda.esquerdo@sjd.es (E.E.); carmen.villalon@sjd.es (C.V.); anna.marin@sjd.es (A.M.); andrea.aldemira@sjd.es (A.A.); 2Pediatrics Department, Hospital Sant Joan de Déu, 08950 Barcelona, Spain; mariona.fernandez@sjd.es; 3Environment Effects on Child/Adolescent Well-Being Research Group, Institut de Recerca Sant Joan de Déu, 08950 Barcelona, Spain; 4Department of Computer Science, Applied Mathematics and Statistics, Universitat de Girona, 17004 Girona, Spain; pepus@imae.udg.edu; 5Faculty of Medicine and Health Sciences, Universitat de Barcelona, 08007 Barcelona, Spain

**Keywords:** pediatric, hospital at home, workload, direct care, training time, visiting time

## Abstract

**Highlights:**

**What are the main findings?**
Patient age, referral specialty, care at home, healthcare team, and complexity are the variables that intervene in the direct workload of acute pediatric hospital at home professionals.

**What are the implications of the main findings?**
Families’ involvement in pediatric hospital at home care facilitates the daily healthcare team workload.It is possible to estimate training time and visiting time according to the above-mentioned variables.

**Abstract:**

**Background:** Hospital-at-home (HAH) programs deliver hospital-level care in patients’ homes, improving satisfaction and optimizing resource use. While widely adopted in adults, pediatric HAH remains limited. At Sant Joan de Déu Hospital (Barcelona, Spain), the pediatric HAH program for acutely ill children has expanded to include more clinically complex cases. Family involvement is essential, as caregivers are trained to administer treatments, monitor clinical status, and support telematic follow-ups, supporting the healthcare team’s workload. **Objective:** To identify patient characteristics influencing healthcare workload and develop a predictive model to enhance resource allocation. **Methods:** This single-center, prospective cohort study included all patients admitted to the pediatric HAH program for one year. Primary variables were caregiver training time, home visiting time, patient age, type of care provided, and clinical complexity. Secondary variables included referral specialty and team composition. Data were collected using digital time-tracking, manual records, and clinical databases. Analyses included Kruskal–Wallis and Dunn’s multiple comparison tests. **Results:** All variables showed significant differences in training and visiting times. Training time ranged from 19 to 157 min; visiting time from 6.2 to 157 min. A predictive model using five key variables estimated visiting time, and another model based on care type estimated training time were created. **Conclusions:** Patient characteristics and caregiver involvement significantly affect direct care workload. These findings can inform strategies to optimize staffing and scale pediatric HAH programs effectively.

## 1. Introduction

Hospital at home (HAH) is a model of care that provides hospital-level treatment in the patient’s home. This approach has been proven to be satisfactory for patients and families [1,2,3] and efficient for the healthcare systems [4], and it also helps free up hospital beds for patients requiring in-hospital infrastructure.

Although HAH is widely implemented across the globe, it is primarily designed for adult populations. In contrast, pediatric HAH remains unusual and often limited to palliative and chronic complex care [1,5]. To date, there are only a few programs dedicated to acutely ill pediatric patients.

In 2019, Sant Joan de Déu Hospital in Barcelona (Spain) launched a HAH specifically addressed to acutely ill pediatric patients [6,7]. Since then, the program has been expanding, incorporating more clinically complex patients. Currently, three primary categories of patients can be identified within the program: (1) respiratory conditions, (2) pathologies requiring intravenous (IV) treatments, and (3) early discharge of complex patients.

Each patient needs an individualized care plan in the home setting. Because the healthcare team usually performs a single visit per day, family involvement is essential in the delivery of care. Specialized nurses in HAH play a key role, providing training in administration of treatments and recognition of potential warning signs. Given the typically brief hospital stays among pediatric patients with acute conditions (4 days mean) [4], it is essential to educate families and enable same-day discharge to home upon referral to HAH services.

Following appropriate training, patients are discharged to their homes to complete their hospitalization. The healthcare team provides home visits and offers support to caregivers as needed. Outside of standard working hours, continuous care is maintained by the Emergency Department.

Routine care includes: (1) in-person consultation to assess the patient’s clinical status, provide reassurance to families, and deliver medication; and (2) two daily telematic consultations—one in the early morning and another in the afternoon—aimed at reassuring families and detecting early signs of clinical deterioration. Additionally, patients requiring IV therapies receive an extra telematic consultation every 48 h to ensure proper medication administration. Exceptionally, some patients may require an additional in-person consultation during periods of clinical instability.

Family involvement is central to pediatric HAH models, and caregiver capacity (skills, resilience, and coping resources) strongly influences care delivery and outcomes [8]. Systematic reviews highlight that caregiver resilience and sense of coherence are associated with better adaptation and lower burden in adult contexts [9]. Similarly, studies in pediatric settings highlight the risk of caregiver burden, both during acute admissions and in the management of chronically complex patients, underscoring the importance of supporting caregiver capacity [7,10]. Recent evidence also underscores that resilience plays a key role in parents’ ability to safely administer medications at home and manage complex care routines [11].

From a resource management point of view, a key challenge involves balancing healthcare personnel’s workload, given that patient-specific characteristics directly influence the time and effort required per case. Consequently, the duration of caregiver training largely depends on the complexity of care required in the home setting. Similarly, the length of the home visit, the need for additional telematic consultations, or the involvement of in-hospital specialists are influenced by the patient’s clinical characteristics and the level of care needed. These activities fall under direct care time [12,13], as they involve face-to-face or direct interaction with the patient. All these factors collectively determine the direct care workload per patient for the healthcare team. In parallel, indirect care time also represents a considerable burden for healthcare personnel [14]. It encompasses tasks that, while essential, do not involve direct patient interaction—such as traveling between locations, reviewing clinical records prior to the visit, and documenting care afterwards. Together, direct and indirect care activities provide a more comprehensive picture of the time investment required per patient and highlight the complexity of managing healthcare resources efficiently.

In the literature, several studies have attempted to quantify and describe the workload associated with HAH programs, focusing primarily on direct care time [12,13,14]. However, these studies have been conducted exclusively in adult populations. To date, there is a lack of published data addressing the specific characteristics and workload implications of pediatric HAH, despite its unique challenges and care dynamics. This gap underscores the need for further research adapted to the pediatric context.

This study aims to identify the key patient characteristics that influence direct healthcare workload and to develop a patient classification framework to support more efficient allocation of healthcare resources in pediatric HAH programs.

## 2. Materials and Methods

### 2.1. Participants

The study included all patients admitted to our HAH program between September 2024 and September 2025. Therefore, no a priori sample size calculation or power analysis was performed, as the sample represented the entire population of HAH patients treated during the study period rather than a selected subset. All patients admitted to the HAH program during the study period were included. No exclusion criteria were applied.

### 2.2. Experimental Design

This was a single-center, prospective study involving a single cohort of patients, in which several variables were assessed within the same patient population.

An admission was defined as a single episode of care in the Hospital-at-Home (HAH) program, beginning at the time of hospital discharge and initiation of home-based treatment and ending when hospital-at-home care was completed or the patient was readmitted to inpatient care. Each admission usually involved several home visits conducted by the healthcare team.

The main outcome variables included: (1) Training time: Defined as the time required by the nurse to prepare materials for home hospitalization and conduct caregiver training; (2) Visiting time: Defined as the total time from the moment the healthcare team parked the vehicle upon arrival at the patient’s home, the duration of the home visit itself, and the return to the vehicle; (3) Patient age: categorized as (a) <1 month old, (b) 1–12 months old, (c) 12–36 months old, (d) 3–6 years old, (e) 6–12 years old, (f) >12 years old; (4) Type of care administered at home, categorized as: (a) Observation, (b) Health education, (c) Infusions of platelets, antivirals, antifungals, or chemotherapy, (d) Intravenous (IV) antibiotic treatment with self-administration, (e) IV antibiotic treatment with nurse administration, (f) Intramuscular antibiotic treatment, (g) Oxygen therapy and inhalation treatments, (h) IV continuous infusion; and (5) Clinical complexity: Defined as the presence of severe chronic conditions, functional limitations, high healthcare utilization, and the need for coordinated, multidisciplinary care. It includes children with neurological, genetic, metabolic, or multisystemic diseases, as well as those with oncological conditions, who also present complex and prolonged care needs.

Secondary outcome variables included: (1) Referral specialty, categorized as: (a) Pediatrics, (b) Onco-hematology, (c) Surgery, (d) Other; (2) Healthcare team composition: (a) One nurse and one physician, (b) Two nurses.

### 2.3. Data Collection Tools

Visiting time was recorded using Routal.com (https://www.routal.com), a route optimization software routinely used by the team. The software captures visit start and end times through the professionals’ secure login activity, ensuring objective and consistent time measurement. Although no specific validation study has been published for this tool, it is fully integrated into the hospital’s electronic workflow system and undergoes regular internal verification to ensure data accuracy and reliability. The company complies with SOC 2 Type II security standards, ensuring platform security and data integrity.

Training time was measured using a stopwatch and manually recorded in an Excel spreadsheet for subsequent analysis.

To minimize bias in manual data collection, training time was consistently recorded by a single nurse per day, using a standardized procedure and stopwatch. Although different nurses participated in data collection across the study period, only one nurse was responsible for recording training time per day, ensuring consistency in each session. Because the nurse in charge on any given day registered all training sessions performed, the process did not involve any selection of patients or cases. Therefore, while some training sessions were not documented due to occasional omissions, this missingness is likely random and unlikely to introduce systematic selection bias into the dataset.

It is important to acknowledge the potential influence of the Hawthorne effect, whereby individuals may alter their behavior simply because they are aware they are being observed or measured. In our study, this could have affected the manual recording of training time, as nurses might have unconsciously adjusted their focus during the task. While this is a recognized limitation of observational studies, we attempted to mitigate its impact by using a standardized and consistent recording procedure.

The procedure was conducted as follows: the measurement began when the nurse started preparing materials for home hospitalization and ended upon completion of caregiver training. No other tasks were performed during this period to ensure accuracy. If an urgent situation arose requiring the nurse’s attention, the stopwatch was paused and restarted once the training task resumed. This protocol was followed to minimize bias and ensure consistency across all recorded cases.

Finally, all data—including clinical variables and exported visit duration data—were compiled into an Excel spreadsheet for centralized storage and analysis.

### 2.4. Handling of Datasets

Two datasets were generated from the same patient cohort, one to analyze the training time, and the other one to analyze the visiting time. The training time dataset included 189 patients, that were collected from the training sessions. The visiting time dataset included 1333 visits from 480 patients and was derived from routine time-stamping records of home visits. Both datasets belonged to the same single-center cohort; however, analyses were performed separately for each dataset.

Both datasets—training time and visiting time—were derived from the same cohort of patients included in the study. The difference in dataset sizes is explained by the nature and recording method of each variable.

Training time was recorded once per admission, as it referred to the nurse-led preparation and caregiver education performed before initiating home hospitalization. In contrast, visiting time was recorded for each home visit, since patients were generally visited once per day throughout the hospitalization period.

Data collection procedures also differed between the two datasets. Visiting time was automatically captured by the Routal software system, ensuring complete digital registration. In contrast, training time was recorded manually by nurses using a standardized procedure and a stopwatch. Only one nurse per day was responsible for documenting all training sessions, and this role rotated among team members to ensure consistency. Because the nurse in charge on any given day recorded all sessions performed, the process did not involve any selection of patients or cases. However, some training sessions were not documented due to occasional omissions, leading to a smaller dataset (n = 189). This missingness is considered random and unlikely to have introduced systematic selection bias.

### 2.5. Statistical Analysis

Data were collected in an internal database. Descriptive statistics included minimum, maximum, mean, median, standard deviation (SD), and interquartile range (IQR). Comparative analyses between variables were conducted.

Since data did not meet the assumption of simple ANOVA, non-parametric tests were applied. Group differences were assessed using the Kruskal–Wallis test, followed by Dunn’s multiple comparison test with the Benjamini–Hochberg adjustment to account for multiple testing. A compact letter display was used to summarize pairwise comparisons: groups that do not share a letter are considered significantly different at the 5% significance level according to Dunn’s test.

For certain categorical variables, groups with low frequencies were excluded from statistical comparisons to avoid sparse data bias.

All statistical analyses were conducted using R (version 4.5.1, R Foundation for Statistical Computing, Vienna, Austria, https://www.R-project.org (accessed on 8 October 2025).

A predictive model was constructed to estimate visiting time using the categorical variables described above (Age group, Referral specialty, Care administered at home, Healthcare team and Complexity).

## 3. Results

### 3.1. Descriptive Analyses of Datasets

A total of 189 records were included in the training time dataset and 1333 records in the visiting time dataset. The distribution of patient age, referral specialty, and type of care administered at home was similar across both datasets. Most patients were between 1 month and 12 years of age. Pediatrics was the most frequent referral specialty. The most common types of care administered at home were oxygen therapy and inhalation treatments, followed by IV antibiotic treatment with self-administration. Further details are provided in Table 1 and Table 2.

In the training time dataset, values ranged from 19 to 157 min, showing high variability (SD = 27.2 min). However, the central 50% of values were concentrated between 45 and 78 min. The distribution was clearly right skewed, with a mean (65.35 min) higher than the median (60 min).

In the visiting time dataset, values ranged from 6.2 to 165 min, again showing high variability (SD = 15.2 min). The central 50% of values were concentrated between 14.9 and 26 min. As observed in the training time dataset, the distribution was right-skewed, with a mean (23.3 min) higher than the median (19.1 min).

### 3.2. Training Time Dataset: Description, Graphical Analysis and Multiple Comparisons Across Categorical Variables

Training time was analyzed across the main categorical variables of interest: Age group, Referral specialty, and Care administered at home. Each variable is presented in the following subsections, including descriptive statistics, graphical representations, and results from multiple comparison analyses, as well as clinical interpretation of the data.

To minimize sparse-data bias, categories with very low frequencies were excluded from the statistical comparisons. Specifically, the <1 month group (n = 4) within Age group, and the Infusions of platelets/antivirals/antifungals/chemotherapy (n = 2) and intramuscular antibiotic (n = 3) categories within Care administered at home were removed from inferential analyses.

Table 3 provides an overview of the distribution of training time across all studied variables and Figure 1 includes the boxplots for each of the variables studied with compact letter display included.

#### 3.2.1. Training Time Across Age Group

Training time increased with patient age, with the highest median values observed in the >12 years group.

The boxplot of training time by age group is presented in Figure 1, illustrating the distribution of the variable. A slight increase in training time with age is apparent, along with greater variability among older patients. Outliers were present in most categories. The Kruskal–Wallis test confirmed significant differences between age groups (χ^2^ = 16.695, df = 4, *p* = 0.002). Pairwise comparisons were performed using Dunn’s test with Benjamini–Hochberg adjustment for multiple testing. A compact letter display was used to summarize the results, indicating which categories differ significantly from each other.

The full results of the Dunn’s post hoc comparisons are presented in Appendix A, Table A1.

These findings suggest that caregivers take on more care-related tasks as children grow older, whereas neonates and young infants typically require a greater degree of direct nursing support in the home setting.

#### 3.2.2. Training Time Across Referral Specialty

Training time differed by referral specialty, with higher mean and median values observed for Surgery and other specialties (mean 88.1 min, median 79 min) and Onco-hematology (mean 89.6 min, median 94 min) compared to general Pediatrics (mean 62.1 min, median 59 min).

The boxplot of training time across referral specialty is presented in Figure 1, where the descriptive analysis above explained is graphically described. In this figure, Surgery and Onco-hematology exhibited comparable distribution patterns. Pediatrics showed generally lower training times, although its outliers were higher in magnitude. The Kruskal–Wallis test confirmed these differences (χ^2^ = 13.835, df = 2, *p* < 0.001). The full results of the Dunn’s post hoc comparisons are presented in Appendix A, Table A2.

These data suggest that training time tends to be longer for patients from specialized referral sources compared to general Pediatrics, although the small sample sizes in the Surgery and Onco-hematology groups should be noted.

#### 3.2.3. Training Time Across Care Administered at Home

Training was longest for tasks requiring greater family involvement, such as IV antibiotics with self-administration (mean 94.2 min, median 90 min) and IV continuous infusion (mean 87.3 min, median 80 min). In contrast, care that was simpler for caregivers to perform, such as Observation (mean 45.4 min, median 38.5 min) and Oxygen/inhalation therapy (mean 58.5 min, median 57 min), required less training. IV antibiotics administered by nurses (mean 62.4 min, median 62.5 min) fell in between, reflecting limited caregiver participation.

Figure 1 presents the corresponding boxplots and highlights three main patterns from the pairwise multiple comparisons. Three patterns were obtained with the pairwise multiple comparisons. IV antibiotics with self-administration and IV continuous infusion—both of which require substantial caregiver involvement—showed the highest median training times, with values that were broadly comparable between the two types of care. IV antibiotics with nurse administration and oxygen therapy/inhalations showed comparable training times, with lower medians, and IV antibiotics with nurse administration and observation were also similar. All of these categories involve procedures that are generally easier and quicker to explain to caregivers. The Kruskal–Wallis test confirmed these differences (χ^2^ = 46.004, df = 4, *p* < 0.001). The full results of the Dunn’s post hoc comparisons are presented in Appendix A, Table A3.

These differences likely reflect the complexity of the procedures: tasks requiring active family participation and technical skill demand more detailed instruction, demonstration, and supervised practice, whereas simpler or nurse-led interventions can be explained more quickly.

### 3.3. Visiting Time Dataset: Description, Graphical Analysis and Multiple Comparisons Across Categorical Variables

Visiting time was examined across the categorical variables. In the following subsections, the variables Age group, Referral specialty, Care administered at home, Healthcare team composition and Complexity are described and compared separately, including descriptive statistics, graphical representations, corresponding multiple comparison analyses, and clinical interpretation of the data.

Categories with low frequencies were excluded from the analysis to avoid sparse data bias. Specifically, the Gastroenterology category (n = 4) from the Referral specialty variable, and the “Infusions of platelets, antivirals, antifungals, and chemotherapy” (n = 10) and Health education (n = 14) categories from the Care administered at home variable were excluded.

Table 4 offers an overview of the distribution of visiting time across all studied variables and Figure 2 includes the boxplots for each of the variables studied with compact letter display included.

#### 3.3.1. Visiting Time Across Age Group

Overall, visiting time tends to be higher in the youngest patients and gradually decreases with age.

The <1 month group has the highest average visiting time (mean 53.3 min, median 52.4 min) and variability (SD 29.9 min), indicating that newborns typically require longer visits. In contrast, from 1 to 12 months onward, visiting times drop substantially, with mean values ranging from 26.0 min in the 1–12 months group to approximately 22 min in children aged 12 months to 12 years. Medians across the older age groups are tightly clustered around 18–20 min, suggesting more standardized and predictable care needs.

Kruskal–Wallis chi-squared confirms the differences: (χ^2^ = 39.997, df = 5, and a *p*-value < 0.001). The full results of the Dunn’s post hoc comparisons are presented in Appendix A, Table A4.

This slight decrease in visiting time with increasing age is stated with the pairwise multiple comparison, shown in the boxplot (Figure 2). Compact letter display confirm these differences with three different ordered patterns obtained with the pairwise multiple comparisons.

This pattern reflects the greater complexity and fragility of younger infants, who typically require more time-intensive assessments, close monitoring, and parental guidance, as well as less active caregiver participation in providing care at home during the admission. In contrast, as children grow older, clinical evaluations and procedures tend to be more straightforward, and families are generally more capable of contributing to routine HAH care. Consequently, visit times become shorter and less variable in the older age groups.

#### 3.3.2. Visiting Time Across Referral Specialty

Onco-hematology visits had the longest durations (mean 28.5 min; median 24.3 min), with the widest variability, reflecting the greater clinical complexity of these patients. In contrast, visits for Pediatrics and Surgery showed similar and shorter durations (means 22.3 and 22.0 min; medians 18.6 and 17.8 min, respectively), consistent with less intensive assessments and more standardized care routines in these groups. Although the Pediatrics group displayed the lowest minimum and median visiting times, it also showed a high number of atypical values.

The Kruskal–Wallis test confirmed significant differences between groups (χ^2^ = 37.664, df = 2, *p* < 0.001). Pairwise multiple comparisons yielded two statistically distinct clusters, represented in the boxplot (Figure 2) using compact letter display: Surgery and Pediatrics versus Onco-hematology. The full results of the Dunn’s post hoc comparisons are presented in Appendix A, Table A5.

This pattern reflects that Onco-hematology patients typically present with greater clinical complexity, requiring more detailed assessments and closer monitoring. This naturally results in longer and more variable visiting times. In contrast, visits for general Pediatrics and Surgery often involve more routine evaluations and stable clinical scenarios, leading to shorter and more consistent visit durations.

#### 3.3.3. Visiting Time Across Care Administered at Home

The longest visits corresponded to IV antibiotics administered by nurses (median 25.3 min), reflecting the need for on-site preparation and administration. Observation visits also showed relatively long and variable durations. Intermediate times were observed for IV continuous infusions, IM antibiotics, and IV antibiotics with self-administration, which generally involve more straightforward or caregiver-led procedures. Oxygen therapy and inhalations had the shortest visiting times (median 17.3 min), consistent with their standardized and quickly performed routines. The Observation category also showed relatively long and variable durations, likely due to the heterogeneity of clinical situations included in this category. The Kruskal–Wallis test confirmed significant differences among groups (χ^2^ = 106.5, df = 5, *p* < 0.001). The pairwise multiple comparisons revealed three distinct patterns (see Figure 2): IV antibiotics administered by nurses formed one group, oxygen therapy/inhalations clustered together with IM antibiotics, and the remaining care modalities comprised an intermediate pattern. The full results of the Dunn’s post hoc comparisons are presented in Appendix A, Table A6.

Overall, visit duration appears to reflect the technical complexity and degree of professional involvement required for each type of care.

#### 3.3.4. Visiting Time Across Healthcare Team Composition

Visits conducted by two nurses had slightly shorter durations (mean 21.0 min; median 16.9 min) compared with visits conducted by a nurse–physician team (mean 23.8 min; median 19.6 min). Variability was also higher in the mixed nurse–physician group. When applying the Kruskall-Wallis test, median times were significantly different (χ^2^ = 19.191, df = 1, *p* < 0.001). Figure 2 shows a boxplot representing the two different categories.

These findings suggest that visits involving a physician may be more comprehensive or require additional assessment time. Based on our experience, this difference is not related to patient complexity, as most patients—regardless of complexity—are visited by two nurses during weekends.

#### 3.3.5. Visiting Time Across Complexity

Patients classified as complex exhibited longer visiting times (mean 29.3 min and median 23.5 min) and greater variability. Kruskal–Wallis chi-squared test showed significant differences (χ^2^ = 79.636, df = 1, *p* < 0.001). The graphical representation of this variable is in Figure 2.

As expected, greater clinical complexity is associated with more intensive and variable care requirements during home visits.

### 3.4. A Classification System for Daily Operations

To optimize the organization of daily home-hospitalization activities, it is essential to estimate the expected duration of the caregivers’ trainings, as well as the home visits, based on patient and care characteristics. A prediction system can help allocate workload more efficiently and anticipate operational needs. In this section, we outline the development of a predictive model aimed at estimating visiting time and a classification to estimate training time.

#### 3.4.1. Development of a Predictive Model for Visiting Time

To support workload planning, we developed a multivariable predictive model for visiting time using a mixed-effects regression framework. Because visiting time exhibited right-skewness, the dependent variable was log-transformed to better meet the assumptions of residual normality and homoscedasticity.

The initial model specification included all clinically relevant predictors:$$\log\left(\mathrm{time}\right)∼\mathrm{Care}+\mathrm{AgeGroup}+\mathrm{Specialty}+\mathrm{Complexity}+\mathrm{TeamComposition}+(1|\mathrm{PatientId}).$$

In this model, not all predictors showed statistically significant associations with visiting time. In particular, Specialty did not contribute meaningfully to the model (*p* = 0.434). Therefore, a reduced model excluding this variable was evaluated:log(time)∼Care+AgeGroup+Complexity+TeamComposition+(1∣PatientId).

A model comparison using a likelihood ratio test indicated no significant difference between the two specifications (ANOVA *p* = 0.433), while the simplified model yielded a lower BIC (1532.7 vs. 1545.3), indicating better parsimony without loss of explanatory value. Consequently, the reduced model was selected. The final regression results are presented in Table 5.

Although the model explains a modest proportion of variance in visiting time (conditional R^2^ = 32.7%), all diagnostic checks indicate that the model is statistically appropriate. The assumptions of normality and homoscedasticity of residuals were met, no problematic multicollinearity was detected (maximum VIF = 2.03), and no influential observations were identified. Posterior predictive checks showed that model predictions closely resembled the observed data. The multicollinearity analysis and the graphical analysis of residuals are attached in Appendix B, Table A7 and Figure A1, respectively.

The estimated baseline visiting time for the reference categories (Neonates; IV antibiotics with nurse administration; Two nurses; Complex patient was exp (3.68) = 39.6 min (SE = 1.1 min). Adjustments for other patients and care characteristics should be applied as shown in Table 5.

As an example, the estimated visiting time for a 4-year-old patient receiving oxygen therapy, referred from Pediatrics, visited by a nurse–physician team, and classified as non-complex is calculated as follows:

The linear predictor is obtained by adding the intercept and the corresponding regression coefficients:3.68 − 0.35 − 0.59 + 0.14 = 2.88

Thus, the model estimates a log-time of 2.88 (SE = 0.0335). Back-transforming this value yields:$$\mathrm{Estimated}\;\mathrm{time}=e^{2.88}=17.8\;min$$

Accordingly, the predicted visiting time for this patient is approximately 17.8 min (95% CI: 16.6–18.9 min). This example demonstrates how visiting time is influenced by patient- and care-specific characteristics. In this particular scenario, the estimated duration is substantially shorter than the model intercept, which represents the reference category including a neonate—patients who typically require longer and more intensive home visits.

By quantifying how visit duration varies according to age group, care type, team composition, and clinical complexity, the model provides a basis for anticipating the expected time required for each visit. For example, the model indicates that neonates generally require substantially longer visits; therefore, assigning several neonates to the same route may lead to an excessive workload. Conversely, visits to older children—for instance, patients aged 3–12 years receiving oxygen therapy—are expected to be shorter, allowing a greater number of patients to be scheduled within the same route.

#### 3.4.2. Classifying Training Time per Care Administered at Home

To facilitate practical application of a simple classification for adjusting the time nurses require to train caregivers in home care, a key variable was selected: care administered at home. This variable represents the specific care that nurses need to teach, making it the most relevant factor for determining training duration.

In Table 6, we propose assigning a training time to each type of care, based on the median values presented in Table 3. Although we recognize that other variables may influence training time, we accept the potential margin of error this approach may entail.

## 4. Discussion

From a healthcare management perspective, our study sheds light on the distribution and determinants of workload within a pediatric hospital-at-home program. Unlike adult HAH models, where workload data has been more extensively reported, pediatric HAH remains poorly documented despite its growing implementation.

We acknowledge that our results are specific to the characteristics of our own setting. Healthcare systems similar to the Spanish model, which provide universal coverage, are likely to encounter comparable patient profiles in pediatric home hospitalization. However, organizational differences—such as whether units allow self-administration of care, implement structured caregiver training, incorporate telemedicine, or deliver care exclusively by nurses versus multidisciplinary teams including physicians—can influence program design and delivery. These variations may limit the direct transferability of our findings, and future research should explore how these models adapt in settings with different resource levels, care coordination infrastructures, and family dynamics.

Nonetheless, the underlying principles are broadly applicable to other pediatric HAH units seeking to optimize time management and workload distribution. Our results highlight the substantial impact of patient complexity, age, type of care administered at home, and caregiver training needs on care time. These insights are essential for healthcare planners, as they illustrate the variable intensity of care per patient and its implications for staff allocation, scheduling, and overall service efficiency. Understanding these dynamics represents a crucial step toward developing pediatric-specific workload models that support sustainable and scalable home hospitalization programs.

In adult HAH programs, care intensity is often quantified using the concept of visit rate per patient per day, as patients may require multiple in-person visits depending on their clinical acuity and the design of the service. This metric is widely used in the literature as a proxy for workload, as seen in the studies by Montalto et al. and Regalado de Los Cobos et al., which describe how variations in visit frequency reflect care complexity and resource needs [13,15]. However, this indicator proves less applicable in the pediatric setting. In our program, it is uncommon for a patient to receive more than one in-person consultation per day, regardless of complexity. Instead, the pediatric model emphasizes family empowerment (using a proper training model), early discharge protocols, and scheduled telematic follow-ups, making visit rate an insufficient measure of workload. This underscores the need to develop alternative indicators tailored to pediatric HAH services, which better reflect the nature of care delivery and time investment required per patient.

Moreover, other studies have relied on validated tools to predict or classify workload in home healthcare settings. However, most of these instruments have been developed for adult populations and include parameters that are not applicable to pediatric care. For instance, commonly used tools such as the USC Home Health Patient Classification System [16] place a strong emphasis on adult-specific concerns, including skin integrity, pressure ulcer prevention, and chronic wound management. Similarly, the ONI scale [17] incorporates indicators of social vulnerability often seen in elderly or socially isolated patients, which do not reflect the typical context of pediatric HAH, where family involvement is high. Other instruments, like the Health Status Score [18], were designed within the U.S. healthcare system and are closely tied to agency-based care models, limiting their relevance in public, universally funded systems like that of Spain. These limitations highlight the lack of pediatric-specific workload assessment tools and reinforce the need for models that account for the unique features of child and family-centered home hospitalization. An interesting point of comparison can be drawn from the Community Health Intensity Rating Scale (CHIRS) [19], a tool developed to quantify the nursing workload intensity in adult HAH programs. CHIRS includes factors such as visit frequency and duration, complexity of interventions, involvement of multidisciplinary professionals, and the degree of caregiver support required.

Although existing tools have proven valuable for resource allocation and workload planning in adult healthcare settings, their direct application to pediatrics remains limited. Pediatric care presents distinct characteristics—such as age-dependent variations in clinical needs, a greater reliance on family members for care delivery, and the essential role of structured caregiver education—that are not adequately captured by adult-oriented models. Indeed, studies in adult HAH programs highlight that family involvement and care coordination are major factors for a satisfactory experience [20], underscoring the importance of integrating these factors into pediatric models. Moreover, previously validated instruments are primarily designed around nursing activities and often overlook medical components such as clinical assessments, treatment decision-making, and physician-led follow-up, all of which are integral to multidisciplinary pediatric HAH programs.

While descriptive studies and program evaluations of pediatric HAH exist [8,21,22], explicit quantitative workload models for acute pediatric care remain scarce. The predictive and training-time classifications proposed in this study extend current knowledge by providing an operational framework to estimate staff workload and guide resource allocation within pediatric HAH services.

We propose a complementary classification system to support pediatric home hospitalization (HAH) resource planning. This classification is divided into two subclassifications.

The first model aims to estimate visiting time and is based on the predictive analysis described above. This model is intended to support the efficient organization and allocation of home healthcare visits.

The second subclassification system has been developed to quantify the nursing training time required for each patient prior to admission to HAH. It highlights the critical importance of caregiver education within the pediatric home hospitalization context. Our findings suggest that greater time invested in caregiver training may be associated with shorter subsequent home visits. This pattern was particularly noticeable among neonate patients, who received shorter training sessions yet required longer home visits. Once caregivers become more confident and familiar with care procedures, healthcare teams can focus home visits on clinical assessment and addressing specific concerns rather than performing the full range of care activities. Our results align with the literature, indicating that caregiver psychosocial resources (resilience, sense of coherence) are associated with more effective home care and lower caregiver burden [23]. Integrating brief assessments of caregiver resilience or sense of coherence into admission procedures could help target additional training or support for families at higher risk of needing intensive professional input.

It is also important to note that caregiver training typically lasts about one hour and is performed only once per admission, whereas home visits occur daily. Consequently, investing sufficient time in caregiver training has a cumulative benefit, optimizing time efficiency and reducing overall workload across the hospitalization period.

Together, these tools provide a structured approach to optimizing both clinical resource distribution and caregiver preparedness within pediatric HAH programs. Nonetheless, the proposed models are exploratory and should be interpreted with caution until externally validated in other pediatric HAH settings. Despite this, they may be adaptable to healthcare systems similar to ours and could serve as a practical framework for other pediatric HAH programs aiming to enhance resource management and care efficiency.

Although this study focused on direct care workload—the time invested to caregiver training and home visits—it is important to recognize that indirect care activities also represent a substantial portion of total workload in pediatric HAH programs. Indirect tasks include care coordination, preparation of materials and medications, documentation, telemonitoring, and communication with families and hospital-based professionals. These activities, while not directly observable at the patient’s bedside, are essential to ensure care continuity and safety.

While not analyzed in the present study, it is reasonable to assume that indirect workload correlates with direct care demands, as higher patient volumes and clinical complexity typically require increased communication and coordination efforts.

The findings of this study support the development of dedicated workload models for pediatric hospital-at-home programs, as outlined below.

### Limitations and Future Research

Given the high variability observed across the studied variables, some categories with small n were removed from the predictive analysis to avoid statistical bias. Further studies should be performed with more data.

Future research should focus on validating these classification systems and exploring their impact on healthcare outcomes and operational efficiency in pediatric HAH settings.

This study focused exclusively on direct care workload, measured as caregiver training and home visiting time. Indirect care activities—such as documentation, coordination, telemonitoring, and communication with families—were not recorded. However, indirect workload is expected to increase in parallel with direct care demands, and its inclusion in future analyses would provide a more comprehensive understanding of total workload in pediatric HAH programs.

Certain contextual factors that might influence visit duration—such as weekday versus weekend differences, weather conditions, or team workload—were not captured in this study. Visiting time was defined strictly as the period from when the healthcare team parked at the patient’s home until returning to the vehicle, thereby excluding travel-related influences. Although most caregivers were new to home hospitalization, and prior experience likely had minimal impact, these unmeasured variables may still have introduced some variability. Future studies should incorporate such contextual covariates to refine understanding of their impact on visit duration and care delivery.

## 5. Conclusions

In conclusion, our study highlights the complexity and variability of workload within pediatric hospital-at-home programs, emphasizing the need for specifically tailored resource management strategies. Furthermore, the proposed classification system offers practical value for both nursing staff and physicians, supporting more efficient planning and care delivery. The existing adult-oriented workload assessment tools, while valuable, fall short in capturing the pediatric-specific dynamics (age-specific clinical needs, family involvement, and caregiver education). Therefore, developing dedicated pediatric workload models that integrate clinical complexity, patient age, care administered at home, and caregiver training is essential to ensure efficient staffing, optimal care delivery, and scalability.

## Figures and Tables

**Figure 1 healthcare-13-03066-f001:**
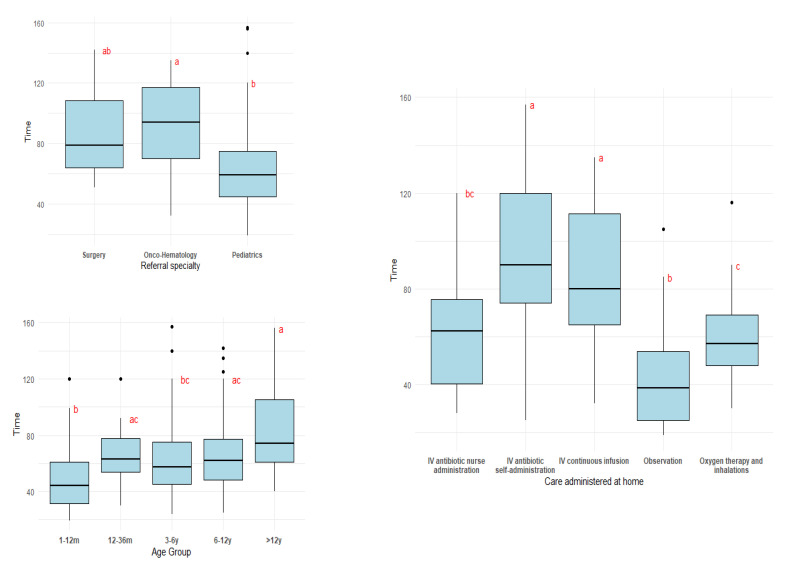
Boxplot showing the distribution of training time (minutes) across the variables studied: Age group, Referral specialty and Care administered at home. Letters (a, b, c) displayed above each box represent the compact letter display, indicating statistically significant differences between groups based on the Dunn test with Benjamini–Hochberg adjustment. Groups that do not share a letter are considered significantly different at the 5% significance level.

**Figure 2 healthcare-13-03066-f002:**
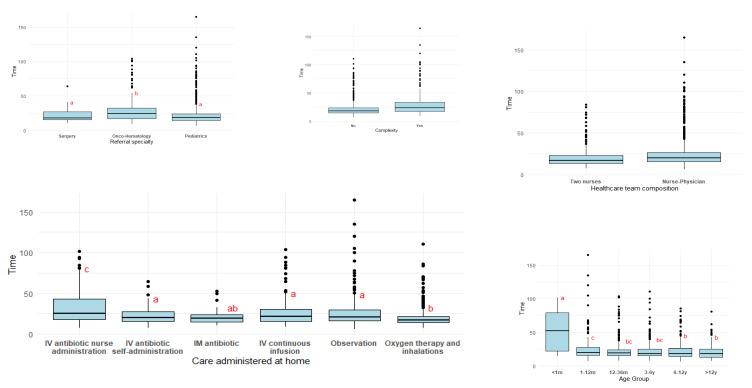
Boxplot showing the distribution of visiting time (minutes) across the variables studied: Age group, Referral specialty, Care administered at home, Healthcare team composition and Complexity. Letters (a, b, c) displayed above each box represent the compact letter display, indicating statistically significant differences between groups based on the Dunn test with Benjamini–Hochberg adjustment. Groups that do not share a letter are considered significantly different at the 5% significance level.

**Table 1 healthcare-13-03066-t001:** Dataset training time. Frequencies of variables of patient age, referral specialty, and care administered at home.

Variable	Categories	N (%)
Patient age	<1 month old	4 (2.1)
	1–12 months old	22 (11.6)
	12–36 months old	41 (21.7)
	3–6 years old	48 (25.4)
	6–12 years old	51 (27)
	>12 years old	23 (12.2)
Referral specialty	Pediatrics	166 (87.8)
	Onco-hematology	16 (8.5)
	Surgery and others	7 (3.7)
Care administered at home	Observation	14 (7.4)
	IM antibiotic	3 (1.6)
	IV antibiotic treatment with nurse administration	18 (9.5)
	IV antibiotic treatment with self-administration	29 (15.3)
	Infusions of platelets, antivirals, antifungals, or chemotherapy	2 (1.1)
	IV continuous infusion	15 (8.0)
	Oxygen therapy and inhalation treatments	108 (57.1)

**Table 2 healthcare-13-03066-t002:** Dataset visiting time. Frequencies of variables of patient age, referral specialty, care administered at home, healthcare team composition, and complexity.

Variable	Categories	n (%)
Patient age	<1 month old	25 (1.9)
	1–12 months old	199 (14.9)
	12–36 months old	363 (27.2)
	3–6 years old	293 (22)
	6–12 years old	306 (23)
	>12 years old	147 (11)
Referral specialty	Pediatrics	1077 (80.8)
	Onco-hematology	214 (16.1)
	Surgery and others	38 (2.8)
	Gastroenterology	4 (0.3)
Care administered at home	Observation	166 (12.4)
	Health education	14 (1.1)
	IM antibiotic	24(1.8)
	IV antibiotic treatment with nurse administration	120 (9)
	IV antibiotic treatment with self-administration	202 (15.1)
	Infusions of platelets, antivirals, antifungals, or chemotherapy	10 (0.8)
	IV continuous infusion	179 (13.4)
	Oxygen therapy and inhalation treatments	618 (46.4)
Healthcare team composition	One nurse and one physician	1081 (81.1)
	Two nurses	252 (18.9)
Complexity	Yes	343 (25.7)
	No	990 (74.3)

**Table 3 healthcare-13-03066-t003:** Descriptive statistics of training time (minutes) across categorical variables Age group, Referral specialty, and Care administered at home. Including mean, standard deviation (SD), minimum (min), first quartile (Q25), median, third quartile (Q75), maximum (max), and sample size (n).

Variable	Mean	sd	Min	Q25	Median	Q75	Max	n
**Age group**								
1–12 months	51.7	26.8	19	31.2	44	60.8	120	22
12–36 months	65.7	17.6	30	54	63	78	120	41
3–6 years	62.1	27.0	24	45	57.5	75	157	48
6–12 years	68.1	28.4	25	48	62	77.5	142	51
>12 years	82.5	31.8	40	61	74	105	156	23
**Referral specialty**								
Surgery and others	88.1	36.3	51	64	79	108	142	7
Onco-hematology	89.6	33.4	32	70	94	117	135	16
Pediatrics	62.1	24.6	19	45	59	75	157	166
**Care administered at home**								
IV antibiotics nurse ^a^	62.4	26.2	28	40.5	62.5	75.5	120	18
IV antibiotics self ^b^	94.2	32.9	25	74	90	120	157	29
IV continuous inf. ^c^	87.3	32.7	32	65	80	112	135	15
Observation	45.4	26.2	19	25	38.5	53.8	105	14
Oxygen/ Inhalations ^d^	58.5	15.5	30	48	57	69	116	108

^a^ IV antibiotics nurse = intravenous antibiotics with nurse administration; ^b^ IV antibiotics self = intravenous antibiotics with self-administration; ^c^ IV continuous inf. = continuous intravenous infusion; ^d^ Oxygen/inhalations = oxygen therapy and inhalation treatments.

**Table 4 healthcare-13-03066-t004:** Descriptive statistics of visiting time (minutes) across categorical variables Age group, Referral specialty, Care administered at home, Healthcare team composition, and Complexity. Including mean, standard deviation (SD), minimum (min), first quartile (Q25), median, third quartile (Q75), maximum (max), and sample size (n).

Variable	Mean	sd	Min	Q25	Median	Q75	Max	n
**Age group**								
<1 month	53.3	29.9	14.4	22.3	52.4	79.1	102	25
1–12 months	26.0	20.2	7.37	15.7	20.2	27.9	165	199
12–36 months	22.5	13.1	7.67	15.4	19.4	24.3	104	363
3–6 years	22.4	13.5	7.83	15.3	18.5	25.0	111	293
6–12 years	21.8	12.2	6.25	14.0	18.3	26.3	85.8	306
>12 years	21.1	11.6	7.57	13.1	18.5	25.1	80.8	147
**Referral specialty**								
Surgery	22.0	10.8	10.5	15.3	17.8	27.1	64.2	38
Onco-hematology	28.5	18.0	8.85	17.0	24.3	32.5	104	214
Pediatrics	22.3	14.6	6.25	14.4	18.6	24.0	165	1077
**Care administered at home**								
IV antibiotics nurse ^a^	33.5	22.4	7.57	18.2	25.3	42.8	102	120
IV antibiotics self ^b^	22.3	9.31	7.37	15.6	20.1	27.6	64.2	202
IM antibiotics ^c^	22	11.5	10.6	14.6	19.1	23.8	52.4	24
IV continuous inf. ^d^	26.3	16.4	8.85	15.9	21.4	30.4	104	179
Observation	27.5	22.6	6.25	16.3	20.7	29.5	165	166
Oxygen/ Inhalations ^e^	19.3	9.91	7.42	13.8	17.3	21.2	111	618
**Healthcare team composition**								
Two nurses	21.0	13.4	7.37	13.6	16.9	22.8	84.2	252
Nurse-Physician	23.8	15.6	6.25	15.2	19.6	26.2	165	1081
**Complexity**								
No	21.2	12.5	6.25	14.2	18.0	23.3	111	990
Yes	29.3	20.1	8.85	17	23.5	33.6	165	343

^a^ IV antibiotics nurse = intravenous antibiotics with nurse administration; ^b^ IV antibiotics self = intravenous antibiotics with self-administration; ^c^ IM antibiotics = intramuscular antibiotics; ^d^ IV continuous inf. = continuous intravenous infusion; ^e^ Oxygen/inhalations = oxygen therapy and inhalation treatments.

**Table 5 healthcare-13-03066-t005:** Regression estimates for predictors of visiting time, including standard errors (std. error) and significance levels.

Mixed Model Complete Results				
	Visiting Time (min)			
Predictors	Estimates	Std. Error	Statistic	*p*
(Intercept)	3.68	0.11	33.29	<0.001
Care IV antibiotic self-administration	−0.17	0.06	−2.92	0.004
Care IM antibiotic	−0.39	0.10	−3.75	<0.001
Care IV continuous infusion	−0.24	0.07	−3.53	<0.001
Care Observation	−0.27	0.06	−4.37	<0.001
Care Oxygen therapy/inhalations	−0.35	0.05	−6.73	<0.001
Age Group 1–12 months	−0.54	0.11	−4.80	<0.001
Age Group 12–36 months	−0.57	0.11	−5.12	<0.001
Age Group 3–6 years	−0.59	0.11	−5.21	<0.001
Age Group 6–12 years	−0.62	0.11	−5.50	<0.001
Age Group >12 years	−0.76	0.12	−6.44	<0.001
Complexity Yes	0.26	0.04	6.23	<0.001
TeamComposition Nurse-Physician	0.14	0.03	4.69	<0.001
**Random Effects**				
σ^2^	0.15			
τ_00_PatientId	0.03			
ICC	0.19			
NPatientId	478			
Observations	1305			
Marginal R^2^/Conditional R^2^	0.173/0.327			

Intercept corresponds to the reference categories: Neonates; IV antibiotic nurse administration; Two nurses; Non-complex patient.

**Table 6 healthcare-13-03066-t006:** Proposed classification for training time.

Care Administered at Home	Estimated Training Time (min)
IV antibiotic treatments with nurse administration	60
IV antibiotic treatments with self-administration	90
Observation	40
IV Continuous infusion	80
Oxygen therapy and inhalation treatments	60

## Data Availability

That data is available upon request because the dataset contains sensitive patient information from pediatric acute home hospitalization cases. Due to ethical and legal constraints, including patient confidentiality and data protection regulations, we are unable to make the full dataset publicly available.

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
