# Peer review of "Predicting Healthcare Workload in Pediatric Home Hospitalization: The Role of Patient Complexity and Family Participation"

_healthcare, 2025, doi:10.3390/healthcare13233066_

Round 1

Reviewer 1 Report

Comments and Suggestions for Authors

I read this manuscript and found it of certain interest. However, it needs revision:  The research design demonstrates merit in its prospective data collection and potential practical applications. 

1. The authors do not provide any data about sample size calculations or power analyses for the dataset (training time: n=189; visiting time: n=1,333). Please clarify the methods for calculating sample sizes that justify the detection of clinically meaningful differences. Also, please make a post-hoc power analysis to determine whether non-significant findings reflect true null effects or inadequate statistical power. You also need to explain the substantial discrepancy between dataset sizes and explain how this may introduce selection bias.

2. We need a flow diagram to clarify how training and visiting records were derived from your cohort. 

3. The inclusion and exclusion criteria should be more clear. 

4. Please add an explanation for "admission" vs. "visit" and the relation between these units of analysis

5. The training times were manually recorded by "a single nurse", raising concerns about inter-rater reliability and observer fatigue.​

6. No information was provided regarding training or standardization of the data collector.

7. Visiting time was utilized by automated software (Routal.com), but no validation of this instrument is presented.

8. You add the "Observation," "Health education," and "IM antibiotic treatment" in the "Other" groups heterogeneous care types without a rationale.​

9. The decision to preserve "Infusions of platelets, antivirals, antifungals, or chemotherapy" as a separate category despite n=2 (training dataset) and n=10 (visiting dataset) violates standard statistical practice for sparse data.

10. With 15 pairwise comparisons for age groups and 15 for care types, more conservative corrections (e.g., Bonferroni-Holm) should be considered.

11. The clinical significance of statistically significant differences is not adequately discussed.

12. The multivariable regression model for visiting time (Table 3) has several weaknesses: Model diagnostics, such as residual plots, multicollinearity assessment, and influence statistics, are not presented. The model R² or adjusted R² is not reported. Finally, no internal or external validation of the predictive model is conducted. The clinical applicability of the model is questionable, given the large standard errors relative to mean visiting times.

13. The manuscript does not address how findings might translate to healthcare systems with different staffing models or family structures.

14. The definition of clinical complexity as "clinical chronic conditions and/or social factors that increase the difficulty of providing care" is vague:.

15 Most of the figures present boxplots without outlier criteria or sample sizes within each category.

Reviewer 2 Report

Comments and Suggestions for Authors

Many thanks to the editor for allowing me to conduct a critical and detailed review of the manuscript entitled “Predicting Healthcare Workload in Pediatric Home Hospitalization: The Role of Patient Complexity and Family Participation.”

Below, I comment on some aspects that I hope will be constructive for improving the manuscript.

  1. Executive summary. In my opinion, the work is novel and relevant: quantifying and predicting the direct healthcare workload (training time and visit time) in a pediatric home hospitalization program. The prospective design and the availability of a large volume of visit records (n = 1,333) are strengths. However, there are methodological and analytical shortcomings that must be addressed before the conclusions can be considered robust and generalizable. Below, I detail the positive points, the main limitations, and the specific improvements required.
  2. Positive aspects. Clinical and organizational relevance. The article responds to a real need for resource planning in pediatric HAH; prospective design and real routine data.

The use of automatic records (Routal) for visiting time brings objectivity to a significant part of the dataset; Sample size for visiting time. 1,333 records is a good basis for descriptive analysis and exploratory modeling of visiting time; Nonparametric analysis and use of post-hoc tests. Kruskal–Wallis and Dunn are appropriate for asymmetric distributions; Practical applicability. Attempt to transform results into operational rules (time classification) useful for managers.

  1. Weaknesses and critical points (methodology and analysis)

3.1. Dependence between observations/repeated data structure

Problem: The visiting time dataset consists of multiple records per patient (average length of stay ~4 days; daily visits). The article treats the 1,333 records as independent observations. This violates the independence of errors and leads to overly optimistic confidence intervals and biased p-values.

Recommendation: Reanalyze using mixed-effects models or generalized estimating equations (GEE) models that include a random intercept per patient (and, if applicable, per professional/shift). This captures intra-patient correlation and produces correct SE and p estimators. Also assess random effects per day or per professional if the structure suggests it.

3.2. Measurement bias: automated vs. manual

Problem: Visiting time is collected automatically (Routal), but training time was measured manually with a stopwatch by a single assigned nurse. These two data sources have different reliabilities and possible biases (Hawthorne, intentional recording). In addition, the training time dataset is much smaller (n=189).

Recommendation: Describe procedures to minimize bias (instructions, recorder training, random verification). Perform a sensitivity analysis to check whether results change for subsamples (e.g., by collection period). Report on quality measures for manual times (duplicates, inter-observer agreement if partially available).

3.3. Predictive model: specification, adjustment, and validation

Problem: The visiting time model presented appears to be a classic multiple regression on records without adjustment for correlation or heteroscedasticity. No adjustment measures (R², RMSE), residual diagnostics, or validation (internal/external) are reported. There is also no explicit control for collinearity.

Recommendation: Adjust mixed models and report: coefficients, robust SE, 95% CI, random effect variance. Report performance measures (e.g., conditional/marginal R² for mixed models, RMSE) and present residual diagnostics (heteroscedasticity, normality/transformation of the outcome if visiting time appears strongly biased). Internal validation: use cross-validation by patients (e.g., leave-one-patient-out or k-fold grouped by patient) or bootstrap to estimate bias and uncertainty and reduce the risk of overfitting. Control multicollinearity (VIF) between concatenated predictors (age, type of care, complexity, specialty). Consider transforming the time variable (log-transform) or modeling it with distributions that account for bias (Gamma/Log-link) if heteroscedasticity persists.

3.4. Grouping/recoding of categories

Problem: The grouping of categories (e.g., types of care, grouped ages) is justified by “practicality,” but the statistical and clinical rationale must be stated and the loss of information quantified. In particular, combining “health education” with invasive techniques in the “Other” category may mask heterogeneity.

Recommendation: Provide a justification table showing original frequencies, clinical criteria for grouping, and compare models with/without grouping to show impact on performance. Avoid groupings that mix levels of clinical complexity without justification.

3.5. Size and power for training time

Problem: n = 189 for training time, with many categories and some very infrequent ones (neonates n=4; QT n=2). Multiple comparisons and modeling on small samples generate unstable results.

Recommendation: Group or exclude categories with very small n for inferential analyses; present descriptive analysis for sparse categories and warn of low power.

3.6. Correction for multiple comparisons and interpretation

The use of Dunn with p-adjust is correct; however, practical interpretation should include effect sizes (medians and differences, not just p). Including CIs for the differences improves clinical utility.

3.7. Confounding and omitted variables

Problem: The possible influence of temporal factors (weekday vs. weekend), distance/travel time, weather conditions, or accumulated workload of the team that day, which can strongly influence visiting time, is not explicitly modeled. Nor is the experience of the caregiver/family (whether there was prior training) or caregiver adherence evaluated.

Recommendation: Incorporate, if available, context covariates (day, time, travel distance, workload on that day). If no data are available, explicitly mention this as a limitation and discuss the possible bias.

3.8. Presentation of results and useful metrics

Recommendation: Add: 95% CI for all estimates, standardized effect measures, and practical calculation examples (one example is already included; expand with intervals). For the proposed model, provide a spreadsheet or algorithm (appendix) to apply the rule in management.

3.9. Generalizability / single-center

Problem: Single-center study in a hospital with a specific organizational model (Spain). Limited generalizability.

Recommendation: Strengthen the discussion on generalizability and the institutional conditions that determine applicability (e.g., universal coverage, nurse/physician ratios, integrated telemedicine).

  1. Are the conclusions supported by the data?

Partially, yes. Descriptive analyses show clear differences in times by age, complexity, team, and type of care. However, the inferential conclusions and predictive model need to be reanalyzed using methods that address intra-patient dependence, internal validation, and diagnostic adjustment. In its current form, the model is at risk of overfitting, and its uncertainty estimates are unreliable for the reasons indicated (non-independence, lack of validation, lack of diagnoses).

  1. Specific recommendations for revision (ordered by priority)

Mandatory revision before re-evaluation:

-Re-estimate visiting time models using a multilevel (mixed-effects) or GEE approach, including patient as a random effect. Repeat diagnosis and report adjustment and validation metrics (conditional/marginal R², RMSE, calibration).

-Internal validation of the model: cross-validation grouped by patient or bootstrap; present out-of-sample performance measures.

-Include residual diagnosis and explore transformations (e.g., log-time, Gamma family) if heteroscedasticity persists.

-Explicitly address repeated data structure and possible temporal or professional bias (add context covariates if available).

-Justify and document recodings/groupings; compare models with/without grouping to show impact.

-Add 95% CI for coefficients and median differences, not just p-values.

-For training time: clarify measurement procedure, discuss limitation of small n in certain categories, and consider robust models or descriptive analysis instead of inferential analysis for sparse categories.

Improvements to presentation and discussion:

Add effect measures (median differences with CI), model performance tables, and calibration/residual graphs. Delve deeper into limitations: manual measurement bias, Hawthorne effect, single-center, absence of data on distance and daily load. Moderate the language of the conclusions: avoid causal terms and present the model as an exploratory tool that requires external validation.

  1. Minor comments and editorial remarks

Add the actual IRB approval (protocol code suggested in the draft) in the corresponding section and detail the management of personal data (anonymization, custody). Clarify in Methods why some types of care (e.g., health education) were grouped with invasive interventions. Include in Supplementary: R code or pseudo-algorithm for prediction (this will facilitate reproducibility for reviewers and readers). Polish minor wording and standardize number of decimal places in tables.

  1. Documentation and literature to add (subtle suggestion of citations)

The work would gain theoretical and practical credibility if it were linked to systematic reviews and studies on caregivers, sense of coherence, resilience, and person-centered assessment tools, given that family involvement is a key variable in the study. I recommend adding them subtly in the introduction and discussion. Including these references will strengthen the argument as to why “family involvement” can modulate workload and will facilitate the connection of your findings with effective interventions for family training and support (and therefore with policy and resource implications). (Suggestion: add these citations in the revised version in the introduction and when discussing the implication of caregiver training and quality of care).

  1. Reviewer's conclusion. The manuscript has merit and potential impact on pediatric healthcare management. However, essential methodological issues (mainly: lack of independence of observations and absence of model validation) must be corrected, and the analysis and presentation must be strengthened before acceptance can be considered. Following methodological revisions and the incorporation of sensitivity analysis and internal validation, the article could become a very valuable contribution to pediatric HAH planning.

Reviewer 3 Report

Comments and Suggestions for Authors

The manuscript addresses an important and timely issue: quantifying and predicting healthcare workload in pediatric home hospitalization (HAH) services. This is a highly relevant topic in modern healthcare management, especially with the increasing adoption of hospital-at-home models. The study employs a prospective design, uses real operational data, and provides a clear effort toward model development. However, the paper has several methodological, structural, and interpretive weaknesses that must be addressed before publication.

  1. The rationale for using non-parametric tests (Kruskal–Wallis and Dunn’s) and the regression approach is not adequately discussed. The authors should explain why more robust modeling techniques (e.g., generalized linear models or mixed-effects models) were not considered.
  2. Although the regression model is presented, no model validation is performed (e.g., residual analysis, cross-validation, or out-of-sample testing). Without such checks, it is unclear whether the model can generalize to future data or different pediatric HAH programs.
  3. The consolidation of multiple categorical variables (lines 169–187) into fewer groups is convenient for operational use, but it may obscure meaningful clinical variation. The authors should present the trade-off between model simplicity and predictive accuracy, ideally with quantitative justification.
  4. Much of the Results section (Sections 3.1–3.4) reads like descriptive reporting rather than analytical depth. For instance, the findings from Figures 7–11 are described narratively but not statistically interpreted beyond visual trends.
  5. The study acknowledges that “limited data in some types of care at home may constitute a bias,” but this requires elaboration. The imbalance between training (n=189) and visiting time (n=1,333) datasets introduces sampling bias. The authors should discuss how this imbalance affects model representativeness.
  6. While the Discussion reviews adult workload models, there is little integration with existing pediatric HAH research.
  7. The regression example is mathematically precise but not conceptually discussed. What does a 33-minute reduction in visiting time for older children mean in practical terms? How can this inform workforce planning or resource allocation? The implications should be discussed in context.
  8. The authors repeatedly acknowledge (lines 87–90, 457) the importance of indirect care time, but neither analyze it nor estimate it. At a minimum, an estimate or discussion of how indirect workload correlates with direct workload would provide a more complete picture.
  9. The numerous boxplots are repetitive and add little interpretive value without statistical commentary. Consider merging or moving several to supplementary material while keeping key visualizations in the main text.
  10. Some figures are not actual figures; they are tables.
  11. Please check "Institutional Review Board Statement". You do not declare it.
  12. The term “Other” category is vague. Please specify which clinical cases are included to ensure transparency.
  13. Please use consistent units (minutes or seconds) across all tables and figures.
  14. Ensure that abbreviations are defined at first mention in the main text, not only in the appendix.
  15. Figures 12–14 and the associated tables (Appendix B) should be better referenced in the Results section to guide the reader.
  16. The manuscript should include a clear limitations subsection within the Discussion rather than scattered statements.
Comments on the Quality of English Language

Line 213: “As sown in Figure 1” → “As shown in Figure 1.”

Line 368: “A key variable was selected: care administered at home.” This could be better expressed as “Care administered at home was selected as the key explanatory variable.”

Round 2

Reviewer 1 Report

Comments and Suggestions for Authors

The authors have responded to all points

Reviewer 2 Report

Comments and Suggestions for Authors

Los autores han abordado todas las cuestiones planteadas. Enhorabuena. El artículo está listo para su publicación.

Reviewer 3 Report

Comments and Suggestions for Authors

Acceptable in the present form